# The Occurrence of Anxiety, Depression, and Distress among Professionals Working in Emergency Care

**DOI:** 10.3390/healthcare12050579

**Published:** 2024-03-01

**Authors:** Luca Anna Ferkai, Bence Schiszler, Bálint Bánfai, Attila Pandur, Gergely Gálos, Zsuzsanna Kívés, Dávid Sipos, József Betlehem, Tímea Stromájer-Rácz, Krisztina Deutsch

**Affiliations:** 1Faculty of Health Science, Doctoral School of Health Sciences, University of Pécs, Vörösmarty Street 4, 7621 Pécs, Hungary; 2Faculty of Health Sciences, Institute of Emergency Care, Pedagogy of Health and Nursing Sciences, University of Pécs, Vörösmarty Street 4, 7621 Pécs, Hungary; bence.schiszler@etk.pte.hu (B.S.); balint.banfai@etk.pte.hu (B.B.); jozsef.betlehem@etk.pte.hu (J.B.); krisztina.deutsch@etk.pte.hu (K.D.); 3Faculty of Health Sciences, Institute of Emergency Care, Department of Oxyology and Emergency Care, University of Pécs, Vörösmarty Street 4, 7621 Pécs, Hungary; pandur.attila@pte.hu; 4Clinical Medical Sciences Doctoral School, University of Pécs, Szigeti Str. 12, 7624 Pécs, Hungary; 5Faculty of Health Sciences, Health Insurance Institute, University of Pécs, Vörösmarty M. Str. 3, 7621 Pécs, Hungary; zsuzsa.kives@etk.pte.hu; 6Department of Medical Imaging, Faculty of Health Sciences, University of Pécs, Szent Imre Street 14/B, 7400 Kaposvár, Hungary; david.sipos@etk.pte.hu (D.S.); timea.stromajer-racz@etk.pte.hu (T.S.-R.); 7József Baka Diagnostic, Radiation Oncology, Research and Teaching Center, “Moritz Kaposi” Teaching Hospital, Guba Sándor Street 40, 7400 Kaposvár, Hungary

**Keywords:** emergency care, occupational health, shift work, physical health, mental health

## Abstract

Maintaining mental health is essential for professions with higher stress levels and challenging environments, including emergency specializations. In this study, the occurrence of distress, anxiety, and depression among a group of ambulance and hospital emergency care professionals was assessed (*n* = 202). A cross-sectional, quantitative, descriptive online survey was conducted, including the internationally validated Beck depression inventory (BDI), the perceived stress scale (PSS-14), and the State–Trait Anxiety Inventory (STAI). Statistical analyses involved descriptive statistics, the χ^2^-test, Mann–Whitney U test, Kruskal–Wallis test, Dunn–Bonferroni test, logistic regression (LR), Cramer coefficient (Cramer’s V), Kolmogorov–Smirnov test, and Spearman’s rank correlation coefficient (r_s_). Based on the results, female professionals are more likely to have depressive symptoms (OR = 2.6, 95% CI = 1.3–5.1), perceived stress (OR = 1.2, 95% CI = 1.2–4.1), and anxiety (OR = 2.1, 95% CI = 1.0–4.1) than male professionals. Perceived stress levels decreased proportionally with increasing years spent working in healthcare (OR = 7.4, 95% CI = 7.1–8.3). Extended work shifts of 12 or 24 h increase the risk of perceived stress and anxiety in emergency care workers (*p* = 0.02). Customized stress management interventions are needed to mitigate the amplified mental strain associated with gender, working years, and longer shifts in the emergency care sector to sustain their mental health and well-being.

## 1. Introduction

The escalating incidence of diverse medical conditions, including mental ailments, highlights the pressing need for a comprehensive examination of the factors that contribute to this phenomenon [1,2,3,4]. It is particularly crucial to prioritize managing our physical and mental well-being [5,6,7], especially for professionals in fields that involve high stress levels and challenging conditions, such as healthcare practitioners specializing in emergency medicine [8].

Emergency medicine professionals are expected to perform exceptionally well at all times, often in critical, life-threatening situations. Through continuous training, they must maintain their expertise and possess skills such as creativity, empathy, patience, and the ability to communicate effectively with patients and colleagues [9]. During their duties, they are frequently exposed to psychologically distressing situations and sights [10,11,12]. Moreover, this profession involves direct hazards, including the possibility of injuries, infections, increased physical stress, and violence, which can lead to further complications [13]. Healthcare providers who work with critically ill patients are exposed to heightened emotional, cognitive, and physical pressures, which can result in several adverse outcomes, such as difficulties managing communication and distress [14], burnout syndrome, and conflicts in personal life [15].

The implications of constantly rotating shifts on the well-being of healthcare professionals are significant and concerning. This phenomenon is cumulative, rendering it a crucial factor in the high attrition rates observed in the specialty [16]. Managing the above-mentioned increased exposure to stress requires internal and external factors, which vary by person. Previous studies have predominantly focused on non-emergency care personnel to assess the issue [17,18]. However, it is equally imperative to evaluate this matter from the standpoint of emergency care workers. This approach would enable us to gain a more comprehensive understanding of the situation and facilitate the development of practical solutions. As such, our research seeks to explore the perspective of emergency care workers on the issue in question. By doing so, we aim to provide meaningful insights into the matter and contribute to the existing body of knowledge.

The objective of our study is to examine the incidence and determinants of anxiety, stress, and depression among emergency care workers. Our investigation entails a comparative analysis of the outcomes of hospital and prehospital emergency care workers to determine the areas that are more susceptible to the development of mental illnesses and the factors that require greater attention to prevent them. By conducting this research, we aim to identify the specific sources of vulnerability within emergency care work and develop effective strategies to mitigate their impact.

## 2. Materials and Methods

### 2.1. Study Design

A quantitative, descriptive, and prospective cross-sectional study characterized by convenience sampling was performed. This study was conducted between January and March 2023, involving 202 workers from the Hungarian Ambulance Service or in-hospital emergency care personnel. It was carried out within the emergency departments of three hospitals and seven ambulance stations in Hungary, three centrally situated and four on the periphery.

### 2.2. Research Sample

Participating professionals working in emergency care were involved, intentionally forming two examined groups, one group including ambulance service professionals and the other group including emergency department personnel working in hospitals. Ambulance service professionals were further divided into two subgroups by their work in central or peripheral regions. G*Power 3.1.9.7 software was used to determine the required sample size and statistical power. According to the results, the minimum value of the total sample size was 183 (effect size: 0.3; α err. prob. 0.05; power (1-β err. prob.): 0.9). Considering the minimum sample size required, we aimed to include a nearly equal proportion of professionals from emergency departments and ambulance stations.

The professionals completed an online questionnaire package during the study, including validated and general questionnaires. Each participant was initially informed about the online survey process, and a consent statement was also filled out. Completing the online questionnaire was voluntary, and professionals could discontinue at any point during the survey. The data collection did not involve questions to identify professionals, ensuring their anonymity. The questionnaire was distributed to each emergency care professional via official email. Also, the questionnaire was shared in relevant social media groups of the mentioned specialties. The questionnaire could only be submitted once from a singular IP address to ensure the data’s integrity. Professionals who failed to completely finish the questionnaire were excluded (*n* = 2).

### 2.3. Data Collection and Analysis

Descriptive statistics [mean, minimum, maximum, standard deviation (SD), interquartile range (IQR)], the χ^2^-test, Mann-Whitney U test, Kruskal–Wallis test, and Dunn-Bonferroni test were used to analyze the collected data. The results were considered significant at a 95% confidence interval, with *p* < 0.05 value. Furthermore, the Cramer association coefficient (Cramer’s V) and Spearman’s rank correlation coefficient (r_s_) were applied. We calculated the odds ratios (OR) to examine the dichotomy variables’ independence. Data recording and statistical analyses were performed using IBM SPSS Statistics 26.0 and G*Power 3.1.9.7.

The normality of the sample was tested using the Kolmogorov–Smirnov test. It showed that the values of the variables investigated in this study followed a non-normal distribution (the significance value for the “Z-score” was less than 5%), so non-parametric procedures were used in further analyses. The logistic regression model incorporated a range of variables, including gender, age, marital status, education level, frequency of night shift work, the impact of secondary employment, number of patients per shift, frequency of depressive symptoms, level of perceived stress, and level of anxiety.

### 2.4. Applied Questionnaires

The following questionnaires were applied during the study:

Beck depression inventory, shortened version: With the Beck depression inventory, the symptoms of depression, the severity of the condition, and its change over time can be assessed [19]. The Hungarian-validated shortened version consists of nine statements, with the answer based on a four-level scale [20]. This version has adequate reliability and is applied as an equivalent to the original [21] (coefficient of Cronbach’s alpha: 0.80). When the test is scored, a value of zero to three is assigned for each answer, and then, the total score is compared to a key to determine the severity of the depression. The score ranged from 0 to 30 maximum. The standard cut-off scores were as follows: 0–9: indicates minimal depression; 10–18: indicates mild depression; 19–25: indicates moderate depression; >26: indicates severe depression.

Perceived stress scale (PSS-14): The PSS-14 comprises fourteen terms, measured on a four-level Likert scale. The scale measures how unpredictable, uncontrollable, and overloaded individuals find their life circumstances. The perceived stress scale examines the factors characterizing personal stress perception, such as experienced stressful situations and the sense of loss of control and unpredictability in everyday life. The questions concern non-specific situations, decreasing the impact of cultural and lifestyle factors on the results. The questions refer to one month, lessening the effect of momentary emotional influence. However, the score indicates effectively the long-term changes in chronic stress levels. For specific questions, higher scores indicate a more frequent occurrence of stressful situations or more successful coping. After transcoding the reversed scores, the total score can be calculated with summation, adding up to the global score of the perceived stress. Scores range from 0 to 56 maximum. Based on the Hungarian validation of the perceived stress questionnaire, the following ranges were used: 0–18 points: low-stress level, 19–37 points: moderate stress level, and 38–56 points: high-stress level [22] (coefficient of Cronbach’s alpha: 0.91).

State–Trait Anxiety Inventory (STAI): The STAI can be applied to measure the level of anxiety. It comprises forty terms [23], measured on a four-level Likert scale. The score of the first 20 questions shows the state anxiety level (STAI-S), while the second 20 refer to the propensity to anxiety (STAI-T). The minimum and maximum available scores are 20 and 80 points in both sections. Scores range from 0 to 80 maximum. Based on the Hungarian validation of the State–Trait Anxiety Inventory, the following ranges were used: 0–26 points: low anxiety level, 27–53 points: moderate anxiety level, and 54–80 points: high anxiety level. The terms also have reversed score questions (coefficient of Cronbach’s alpha: 0.91) [24].

In addition, a general questionnaire was also applied, including questions regarding sociodemographic factors, working conditions, leisure activities, personal health interpretation, and personal health behavior. It is essential to tailor questionnaires to the specific sample being studied, so that all relevant factors can be assessed. Therefore, the questionnaire used in this study included a mix of open-ended and closed-ended questions that addressed the unique aspects of the studied sample.

## 3. Results

### 3.1. Sociodemographic Characteristics of the Sample

In the online survey, 202 emergency professionals took part (*n* = 202). The study under consideration involved completing a questionnaire by 125 professionals (61.9%) from ambulance services and 77 emergency department professionals (38.1%) associated with hospitals. The ambulance service professionals were further classified into 60 professionals working in the central region (48%) and 65 working in the periphery area (52%).

Based on the results, males were overrepresented in the sample, and most professionals were middle-aged (ranging between 18 and 63 years; SD: 10.40; IQR: 18.0; Q1–Q3: 28–46). The study involved 74 female professionals (36.6%) and 128 men (63.4%). Most of them graduated from secondary school, with the share of elementary school graduates being low. Nearly all professionals were married or in a relationship, though many were single or divorced. The more significant part of them resided in towns. Regarding sociodemographic characteristics of the professionals by gender, the groups had similar characteristics. Among the female professionals who worked as emergency care professionals, there was a higher proportion of college and university graduates than among males, possibly due to the recruitment criteria for certain work positions. Presently, the representation of female professionals as ambulance drivers or nurses in the Hungarian prehospital care sector is limited. This is mainly due to the absence of a standardized recruitment physical fitness test. As a result, many female professionals who work in the industry tend to occupy roles such as paramedics or emergency doctors, which require tertiary-level education. 

The present study examined the marital status, parental status, and occupational characteristics of emergency care professionals, explicitly concerning gender. Our analysis revealed that most individuals in this cohort were married, while the lowest proportion was divorced. Notably, females had a higher percentage of singles than males. More than half of the female professionals had no children, while a significant portion of the remaining female professionals had one or two children. This trend was also observed in male professionals (Table 1).

Regarding occupational characteristics, most female professionals were employed in emergency departments, whereas males tended to work in prehospital settings. Notably, none of the female professionals held positions as ambulance drivers, while male professionals predominantly served as nurses. Regarding work schedules, female professionals mainly worked 12 h shifts, whereas males were more inclined towards 24 h shifts. Interestingly, most emergency care professionals, irrespective of gender, worked night shifts, and a significant portion held secondary employment. Additionally, this study examined emergency care professionals’ smoking and coffee consumption habits during shifts. It was found that almost all emergency care professionals regularly consumed coffee at the workplace, with a slightly higher proportion observed among female professionals. Most professionals in our study have over ten years of work experience on average, ranging from 0.5 to 41 years (SD: 9.9; IQR: 14; Q1–Q3: 5–19). Emergency care professionals attend to many patients, with varying treatment volumes depending on the location. Patient numbers per shift ranged from one to a hundred (SD: 16.2; IQR: 10; Q1–Q3: 5–15). Typically, central care areas handled higher patient volumes than peripheral areas, with relatively lower patient numbers. However, emergency care professionals spent more time with each patient due to longer transport times for prehospital care in peripheral areas. Additionally, emergency care professionals encountered a notable number of distressing emergency care cases annually, ranging from zero to two hundred (SD: 19.5; IQR: 8; Q1–Q3: 2–10). These characteristics are summarized in Table 2.

Figure 1 displays the distribution of emergency professionals’ leisure activities in our study. Professionals were presented with a multiple-choice question, allowing for the selection of multiple activities. Interestingly, 27 individuals (13.4%) reported engaging in activities beyond those explicitly listed in the questionnaire. These supplementary activities encompassed diverse pursuits such as dancing, video gaming, “do-it-yourself” projects, dog walking, motorcycling and bicycling, hunting, hiking, handicrafts, scale modeling, winemaking, and photography. Analysis revealed that 56 professionals (27.7%) regularly participated in one or two leisure activities, while 96 (47.5%) were engaged in three or four, and 50 (24.8%) reported involvement in more than four types of activities.

This study asked professionals about their medical conditions. Among the respondents, 128 (63.4%) reported known chronic illnesses, with hypertension being the most common (52 cases, 25.7%). Other reported conditions included diabetes, hypercholesterolemia, bronchial asthma, esophageal reflux, hypothyroidism, hyperthyroidism, chronic heart disease, atrial fibrillation, and hyperuricemia.

### 3.2. Summary of Descriptive Results of the Validated Questionnaires

The outcomes revealed that the Beck depression inventory indicated that among the surveyed emergency care professionals, the average total score suggested mild depressive symptoms. Specifically, 23 professionals (11.4%) displayed average scores, 158 (78.2%) exhibited mild depressive symptoms, 20 (9.9%) showed moderate symptoms, and one (0.5%) displayed severe symptoms. 

The results from the PSS-14 scale indicate that the average subjective stress level among emergency care professionals was relatively low, at 37.5%. Most professionals reported not feeling significantly stressed. 

The STAI questionnaire indicated that both the state anxiety level (50%) and the propensity to experience anxiety (48.8%) were moderate based on the total scores. Specifically, the distribution of state anxiety levels among professionals was as follows: low for 110 professionals (54.5%), medium for 24 (11.9%), and high for 68 (33.7%). Regarding the propensity for anxiety, it was low for 93 professionals (46%), medium for 50 (24.8%), and high for 59 (29.2%). The total scores of the validated questionnaires mentioned above can be seen in Table 3.

### 3.3. Detailed Results of the Validated Questionnaires

#### 3.3.1. Beck Depression Inventory (BDI)

There was a significant disparity in the prevalence of depressive symptoms observed between the genders of emergency care professionals (U = 3706.5; *p* = 0.01, Table 4). Our investigation into the risk factors associated with depressive symptoms using logistic regression analysis identified gender as a significant predictor (OR = 2.6, 95% CI = 1.3–5.1; Table 5. This finding suggests that female emergency care workers are 2.6-times more likely to experience depressive symptoms compared to their male counterparts. Moreover, emergency care professionals working 12 or 24 h shifts exhibited a notably higher prevalence of depressive symptoms than those on 8 h shifts (*p* = 0.02). Additionally, our results revealed that professionals who encountered distressing emergency care cases less frequently had a lower prevalence of depressive symptoms compared to those facing such situations more often (*p* = 0.01). These results are shown in Table 4.

In the conducted research, it was observed that the remaining variables under investigation exhibited no statistically significant relationship with the prevalence of depressive symptoms.

#### 3.3.2. Perceived Stress Scale (PSS-14)

Most of the professionals reported not feeling significantly stressed. However, a notable difference in perceived stress levels between genders among emergency care professionals was observed (U = 3329.5; *p* < 0.001; Table 6). Further analysis through logistic regression revealed that female emergency care workers were more likely to experience stress symptoms compared to their male counterparts. Specifically, females were twice as likely to report stress symptoms as males (OR = 1.2, 95% CI = 1.2–4.1). Additionally, there was a significant association between higher perceived stress scores and the likelihood of experiencing depressive symptoms (*p* < 0.001; r_s_ = 0.7; Cramer’s V: 0.69). Newcomer employees in emergency care are over seven-times more likely to experience stress than those with more years of experience in the field (OR = 7.4, 95% CI = 7.1–8.3). These results are summarized in Table 5. 

Among emergency care professionals, divorced individuals reported higher perceived stress levels compared to their married counterparts (*p* = 0.02), while higher qualifications at the workplace were associated with lower perceived stress (*p* = 0.001). Those working 12 and 24 h shifts exhibited higher perceived stress than those on 8 h shifts (*p* = 0.02). Our results revealed that professionals who encountered distressing emergency care cases less frequently had a lower level of perceived stress than those who experienced such situations more than 50 times per year (*p* = 0.04). These results are shown in Table 6.

In this study, it was noted that the other variables examined showed no statistically significant correlation with the perceived stress levels.

#### 3.3.3. The State–Trait Anxiety Inventory (STAI)

The results of the STAI-S questionnaire show that there is a significant disparity in state anxiety levels that emerged between genders among emergency care professionals (U = 3559.0; *p* = 0.003; Table 7). Female emergency care professionals exhibited a higher susceptibility to state anxiety symptoms compared to their male counterparts (OR = 2.1, 95% CI = 1.0–4.1; Table 5. It suggests that female emergency care workers are twice as likely to experience state anxiety symptoms as their male counterparts. Furthermore, the level of state anxiety was found to be higher among single and divorced emergency care professionals compared to those in registered partnerships (*p* = 0.02). Emergency care professionals with three or four children demonstrated lower levels of state anxiety than those without children (*p* = 0.003). Similarly, emergency care workers on 24 h shifts exhibited lower levels of state anxiety compared to those on 12 h shifts (*p* = 0.02). Moreover, emergency care providers who encountered more than ten distressing cases per year reported higher levels of state anxiety than those who experienced no distressing situations (*p* = 0.04). These results are shown in Table 7.

The results of the STAI-T questionnaire showed a notable difference in the level of propensity anxiety between genders among emergency care professionals (U = 3580.0; *p* = 0.004; Table 7). Female professionals in emergency care were nearly twice as likely as their male counterparts to have symptoms of propensity anxiety. Statistical analysis indicated a significant difference in propensity anxiety levels between genders (OR = 1.9, 95% CI = 1.1–3.6; Table 5). Furthermore, divorced emergency care providers showed higher levels of propensity anxiety compared to their married counterparts (*p* = 0.02). Emergency care professionals with three or four children displayed reduced levels of propensity anxiety compared to those without children (*p* = 0.04). Likewise, emergency care workers on 24 h shifts had lower levels of propensity anxiety than those on 12 h shifts (*p* = 0.02). Additionally, emergency care providers encountering more than ten distressing emergency care cases per year demonstrated a higher propensity for anxiety symptoms compared to those who did not have any of them (*p* = 0.02). These results are shown in Table 7.

In the research conducted, it was found that the other variables examined did not demonstrate any statistically significant association with the level of state and trait anxiety.

The influencing factors based on the results of the validated questionnaires mentioned above can be seen in Table 5. This table exclusively presents statistically significant outcomes. Of note, the remaining variables integrated into the model, including age, marital status, education level, frequency of night shift work, the impact of secondary employment, and the number of treated patients per shift, did not demonstrate a statistically significant impact on the results.

## 4. Discussion

Our research revealed that emergency care professionals working 12 or 24 h shifts exhibited a notably higher prevalence of depressive symptoms and perceived stress than those on 8 h shifts, and those who encountered distressing emergency care cases less frequently had a lower prevalence of depressive symptoms compared to those facing such situations more than 50 times per year. This finding is noteworthy as it underscores the association between extended shifts and heightened rates of depressive symptoms among emergency care professionals, which is also reflected in the results of another research [25]. Moreover, this relationship is compounded by an escalation in the perceived mental burden of their tasks. The outcomes of our study reveal that female professionals who work in emergency care exhibit significantly elevated levels of anxiety, stress symptoms, and depressive symptoms compared to their male colleagues. The implications of these findings are worth considering. The observed differences represent statistically significant findings, signifying that female emergency care workers’ likelihood of experiencing these symptoms is markedly increased. These findings emphasize the criticality of addressing the unique stressors and challenges faced by female emergency care providers, as well as the necessity for interventions to improve their mental health and overall well-being. Further research could be undertaken to identify the root causes of these gender-based differences and guide evidence-based policies and practices aimed at supporting the mental health needs of all emergency care workers. According to a previous study, professionals in emergency care face unique challenges, including physical demands, discrimination, and underrepresentation in leadership roles. Healthcare organizations must prioritize inclusiveness, diversity, and equity in the workplace by implementing policies and practices that support female professionals and create a more supportive and inclusive environment [26]. Many types of stress management exercises, such as autogenic training and mindfulness, are widely taught and used. Their positive effects have been proven by numerous professional studies. Regular practice can help improve long-term stress management, quality of life, and the risk of burnout at work [27,28].

Notably, individuals employed in the healthcare sector tend to exhibit lower levels of perceived stress as their tenure in the profession increases. This phenomenon can be attributed to the fact that novice workers are often confronted with unfamiliar experiences, such as caring for unwell or injured patients and being exposed to the workplace’s various sights, sounds, and smells. Furthermore, inexperienced workers are frequently entrusted with immediate patient care responsibilities upon completing their educational training, which may exacerbate their initial apprehension. However, if these stressors remain unmanaged, they may lead to burnout, ultimately resulting in the eventual alleviation of anxiety levels in the end. Mental health prevention programs are crucial to workplace wellness initiatives, particularly for healthcare professionals encountering daily stress. By offering these programs to employees at the onset of their tenure, employers can effectively reduce acute stress levels and mitigate the risk of burnout and associated career loss [29,30]. 

Based on our research, we found that individuals who scored higher on the perceived stress scale were more likely to experience symptoms of depression. These findings underscore the need to proactively identify the presence of stress and take appropriate measures to manage it effectively. Failure to do so may exacerbate the condition and could lead to the onset of depressive symptoms. It highlights the importance of early detection and proper management of stress to prevent the development of more serious mental health issues. Providing appropriate training and support may be critical in helping emergency care professionals better manage the challenges and stressors inherent in this line of work [6].

Effective management of emergency care professionals’ stress levels is a critical concern, as the nature of their work exposes them to essential incidents daily, as highlighted by Donnelly’s research [31]. This concern is further amplified by Afshari’s perspective, which posits that emergency care workers are constantly exposed to occupational stressors that can adversely affect their health [32]. Chronic stress reduction is necessary to ensure employee retention in this sector, a significant factor in this field of work, according to Herttuainen and Nordquist [33]. Khan’s research findings corroborate this assertion by demonstrating that paramedics have a higher incidence of anxiety, depression, regular fatigue, and insomnia compared to the standard population [34]. The findings of these studies corroborate the phenomena and circumstances we have personally encountered. 

Our research showed that state anxiety levels were higher for those who worked 12 or 24 h shifts than those who worked 8 h shifts. According to a study, regular exercise is an effective means of reducing anxiety levels in this population [35]. Although studies on similar topics support these findings, previous research has focused only on nursing staff or health science/medical students, with little attention to the impact of working in shiftwork. A gap in the literature exists regarding the specific experiences of hospital and prehospital emergency care workers concerning their anxiety levels and working schedules, with few studies addressing this topic directly. Thus, the state study aimed to address this gap and provide a more comprehensive understanding. For example, Busa and colleagues studied health science students’ sleep patterns and stress levels. Their results showed that a lack of physical activity was associated with higher stress levels and deteriorated sleep quality [36]. On this basis, regular physical activity can be considered paramount in this field while studying health sciences. Therefore, these problems have already arisen during health studies. Another study found that sleep disturbances are high among emergency care professionals, which are the main symptoms of maladjustment to shift work [37]. A different study emphasized the significance of shift workers not getting enough sleep regularly, which can lead to significant health and well-being issues. The emergency care providers who participated in the study needed a greater understanding of sleep hygiene and its significance. As a result, they believed that more focus should be placed on creating interventions that can improve sleep quality [38]. According to another study to mitigate these adverse effects, it is imperative to thoughtfully design schedules for shift workers that allow for adequate anchor sleep periods. It is essential to minimize daytime responsibilities for these individuals. Consideration of various incentives to compensate professionals working night shifts predominantly may also prove beneficial. It is advisable to provide them with a designated facility for rest before driving home after night shifts [16]. The relationship between sleep and mental health difficulties has been the subject of recent research. Scott et al. conducted a study that found a causal relationship between sleep and mental health [39]. Ghalichi et al. investigated the sleep habits of healthcare workers. They discovered that sleep problems, particularly among female professionals, may be an early indication of specific mental health problems and can impact work performance [40]. Harris noted the prevalence of depression, anxiety, and shift work disorder (SWD) among paramedics, conditions that often co-occur with mental health disorders [25]. These studies highlight the importance of understanding the connection between sleep and mental health and its significant impact on the workplace. It would be worthwhile to add these conditions to our observations in the future to assess their impact in this sample.

Barrea and colleagues studied the effects of coffee consumption on the human body. They found that consuming up to 400 mg of caffeine is safe (1–4 cups daily). However, excessive consumption can cause restlessness, palpitations, irritability, and sleep disturbances [41]. Our study highlighted that a significant proportion of emergency care professionals engage in smoking and coffee consumption during shifts as habitual practices. The impact of caffeine consumption on emergency workers is a subject of interest in academic and professional circles. A study conducted in this regard found that caffeine can positively affect their psychomotor performance and alertness, which is advantageous for their line of work. However, it has also been noted that excessive caffeine intake can negatively affect sleep quality and duration. It underscores the importance of educating emergency workers on responsible coffee consumption habits to ensure optimal performance and overall well-being [42]. Pepłońska and colleagues investigated how working night shifts affects workers’ coffee consumption habits. They observed that those who often worked at night drank more coffee [43]. It suggests that more emphasis should be placed on building emergency care workers’ appropriate coffee consumption and smoking habits.

Vincze et al. investigated the smoking habits of the employees of the Hungarian Ambulance Service. They identified an unfavorable prevalence of smoking and nicotine product use and a high intention to quit among ambulance workers [44]. To meet the dynamic needs of prehospital patient care, regular monitoring of ambulance workers’ smoking and other lifestyle risk behaviors and the design and implementation of targeted workplace health promotion programs would be critical. Health workers are recognized by society for their expertise and are seen as role models for healthy lifestyles, which is why they must set a good example for the public.

Our results show that female emergency care workers are twice as likely to experience state anxiety symptoms as their male counterparts. Furthermore, the level of state anxiety was found to be higher among single and divorced emergency care professionals compared to those in registered partnerships. A study explained this phenomenon by the utilization of multiple perceived mental support types related to stressful emergency scenarios with lower levels of state anxiety [45]. According to Carvello: “Among the strategies aimed at mitigating the effects of this phenomenon, peer-supporting represents an emerging model used in the emergency service setting”. A good way forward is to increase peer support to reduce stress and prevent burnout syndrome [46,47]. It would be advisable to consider testing these variables within our sample.

## 5. Limitations

While this study has provided valuable insights, it is essential to acknowledge its limitations. Firstly, the reliance on self-reported data introduces the possibility of response bias and subjective interpretation, particularly concerning variables such as well-being and stress levels. Additionally, incomplete questionnaire responses due to high stress levels or burnout syndrome among employees may have skewed the results. Given the multifaceted nature of these factors, a comprehensive examination of their effects is warranted.

To address these limitations, future research should aim for larger sample sizes. Longitudinal or experimental designs would offer better clarity on causal pathways. Furthermore, this study overlooked several variables which can have impact the subject matter. Notably, the domestic roles of female professionals were not considered, which could contribute to heightened stress levels and increased incidence of depressive symptoms. The demands of child-rearing and household responsibilities may have substantial effects on stress levels and mental health, meriting exploration in future studies. The female demographic within the emergency care workforce presents an intriguing avenue for inquiry. More than half of the female professionals in this sector do not have children, raising questions about the unique aspects of working in emergency care or other underlying factors. Further investigation into this demographic is essential for a deeper understanding of the phenomenon.

Lastly, it is advisable to explore the potential of leisure activities in reducing stress levels. Further scrutiny of leisure pursuits could shed light on their efficacy in promoting well-being among emergency care professionals.

## 6. Conclusions

Based on the discussion provided, several key conclusions can be drawn. Emergency care professionals working extended shifts (12 or 24 h) exhibit a higher prevalence of depressive symptoms and perceived stress compared to those on shorter shifts (8 h). It suggests an association between extended shifts and heightened rates of depressive symptoms, compounded by increased perceived mental burden.

Female emergency care professionals experience significantly elevated levels of anxiety, stress symptoms, and depressive symptoms compared to their male colleagues. Addressing the unique stressors and challenges faced by female professionals is crucial to improving their mental health and overall well-being. Stress management exercises such as autogenic training and mindfulness have been proven effective in improving long-term stress management and quality of life and reducing the risk of burnout among emergency care professionals. Novice employees in the healthcare sector often experience higher levels of perceived stress due to unfamiliar experiences and responsibilities. However, effective stress management programs can mitigate this risk and prevent burnout. Emergency care professionals working night shifts experience higher levels of anxiety, emphasizing the importance of regular exercise and proper sleep hygiene in managing stress levels. Excessive caffeine consumption, smoking, and irregular eating habits among emergency care professionals can negatively impact their well-being and performance, highlighting the need for education on responsible habits. Female emergency care professionals and those who are single or divorced are more likely to experience anxiety symptoms. Peer support programs can help mitigate stress and prevent burnout among these individuals. Workplace health promotion programs targeting lifestyle risk behaviors such as smoking are essential for maintaining the well-being of emergency care professionals.

In summary, addressing the unique stressors faced by emergency care professionals, mainly female workers and those on extended shifts, through effective stress management interventions and support programs is crucial for promoting their mental health and overall well-being.

## Figures and Tables

**Figure 1 healthcare-12-00579-f001:**
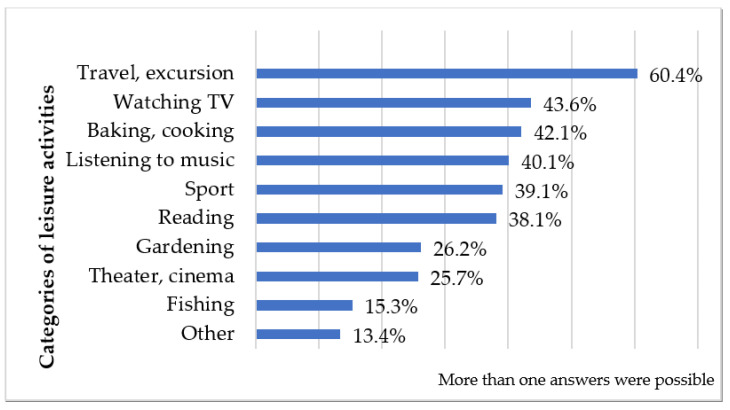
The most common leisure activities of emergency care workers (*n* = 202).

**Table 1 healthcare-12-00579-t001:** Sociodemographic characteristics of the emergency professionals by gender (*n* = 202).

	Female (*n* = 74)	Male (*n* = 128)	Total
Variables	*n* (%)		
Educational attainment			
Elementary school	3 (4.1%)	13 (10.2%)	16 (7.9%)
Secondary school	26 (35.1%)	76 (59.4%)	102 (50.5%)
College/university	45 (60.8%)	39 (30.5%)	84 (41.6%)
Marital status			
Single	14 (20.3%)	25 (19.5%)	39 (19.3%)
Registered partnership	22 (31.9%)	33 (25.8%)	55 (27.2%)
Married	33 (47.8%)	66 (51.6%)	99 (49%)
Divorced	5 (7.2%)	4 (3.1%)	9 (4.5%)
Number of children			
Have no children	38 (51.3%)	53 (41.4%)	91 (45%)
1–2	28 (37.8%)	59 (46.1%)	87 (43.1%)
2–4	7 (9.5%)	16 (12.5%)	23 (11.4%)
>4	1 (1.4%)	-	1 (0.5%)
Type of residence			
Large town	19 (25.7%)	29 (22.7%)	48 (23.8%)
Town	42 (56.8%)	69 (53.9%)	111 (54.9%)
Village	13 (17.6%)	30 (23.4%)	43 (21.3%)

**Table 2 healthcare-12-00579-t002:** Occupational characteristics of the emergency professionals by gender (*n* = 202).

	Female (*n* = 74)	Male (*n* = 128)	Total
Variables	*n* (%)		
Workplace			
Hungarian Ambulance Service	24 (32.4%)	101 (78.9%)	125 (61.9%)
Emergency Department	50 (67.6%)	27 (21.1%)	77 (38.1%)
Qualifications			
Ambulance driver/rescue technician	-	32 (25%)	32 (15.8%)
Emergency nurse	37 (50%)	67 (52.3%)	104 (51.5%)
Paramedic	25 (33.8%)	23 (18%)	48 (23.8%)
Emergency doctor	12 (16.2%)	6 (4.7%)	18 (8.9%)
The type of shift			
8 h	4 (5.4%)	3 (2.3%)	7 (3.5%)
12 h	40 (54.1%)	11 (8.6%)	51 (25.2%)
24 h	16 (21.6%)	63 (49.2%)	79 (39.1%)
Also, 12 and 24 h	14 (18.9%)	51 (39.8%)	65 (32.2%)
Works night shift			
Yes	70 (94.6%)	126 (98.4%)	196 (97%)
No	4 (5.4%)	2 (1.6%)	6 (3%)
Secondary employment			
Yes	33 (44.6%)	87 (68%)	120 (59.4%)
No	41 (55.4%)	41 (32%)	82 (40.6%)
Regular coffee consumption during the shift			
Yes	67 (87%)	105 (82%)	172 (85.1%)
No	10 (13%)	23 (18%)	33 (14.9%)
Regular smoking during the shift			
Yes	29 (39.2%)	51 (39.8%)	80 (39.6%)
No	45 (60.8%)	77 (60.2%)	122 (60.4%)

**Table 3 healthcare-12-00579-t003:** Total scores of the validated questionnaires (*n* = 202).

Total Scores	Min.	Max.	IQR (Q1–Q3)	Mean ± SD
**BDI ^1^**	9	34	4 (11–15)	13 ± 3.66
**PSS-14 ^2^**	2	47	13 (15–28)	21 ± 8.52
**STAI-S ^3^**	20	76	19 (30–49)	39 ± 11.49
**STAI-T ^4^**	20	74	15 (32–47)	40 ± 10.26

^1^ Beck depression inventory; ^2^ perceived stress scale; ^3^ the state anxiety inventory; ^4^ the trait anxiety inventory. Min: minimum; Max.: maximum; IQR: interquartile range; SD: standard deviation.

**Table 4 healthcare-12-00579-t004:** Context analysis of emergency professionals’ Beck depression inventory scores (*n* = 202).

**Variables**	**N**	**Mean Rank**	**Sum of Ranks**	**U**	***p*-Value**
Gender				3706.5	0.01 *
Female	74	115.41	8540.50		
Male	128	93.46	11,962.50		
Workplace				4432.0	0.34
Hungarian Ambulance Service	125	98.46	98.46		
Emergency Department	77	106.44	106.44		
Works night shift				555.5	0.82
Yes	196	101.67	19,926.50		
No	6	96.08	576.50		
Secondary employment				4618.0	0.46
Yes	120	98.98	11,878.00		
No	82	105.18	8625.00		
Regular coffee consumption				2418.0	0.58
Yes	172	102.44	17,620.00		
No	30	96.10	2883.00		
Regular smoking				4375.0	0.21
Yes	80	95.19	7615.00		
No	122	105.64	12,888.00		
Chronic illnesses				4716.0	0.96
Yes	74	101.77	7531.00		
No	128	101.34	12,972.00		
**Variables**	**N**	**Mean Rank**	**Kruskal–Wallis H**	**df**	***p*-value**
Educational attainment			0.531	2	0.77
Elementary school	16	106.50			
Secondary school	102	103.52			
College/university	84	98.09			
Marital status			7.162	3	0.07
Single	39	115.65			
Registered partnership	55	101.91			
Married	99	92.83			
Divorced	9	133.06			
Number of children			1.229	2	0.54
Have no children	140	104.34			
1–2	56	96.00			
3–4	6	86.58			
Type of residence			1.335	2	0.51
Large town	48	107.06			
Town	111	97.23			
Village	43	106.33			
Qualifications			4.394	3	0.22
Ambulance driver/rescue technician	32	87.23			
Emergency nurse	104	104.18			
Paramedic	48	97.83			
Emergency doctor	18	121.17			
The type of shift			9.645	3	0.02 *
8 h	7	84.93			
12 h	51	121.80			
24 h	79	90.45			
Also, 12 and 24 h	65	100.78			
Working years in the healthcare sector			0.881	3	0.83
0–10 years	101	104.61			
11–21 years	58	98.00			
22–30 years	29	102.26			
31–41 years	14	91.96			
Number of treated patients/shift			2.311	3	0.51
1–5	78	97.53			
6–10	64	97.60			
11–20	32	108.42			
>20	28	113.57			
Number of distressing emergency care cases/year			11.026	3	0.01 *
0	10	47.25			
1–10	157	101.83			
11–50	29	113.14			
>50	6	127.08			
Number of leisure activities/person			1.645	2	0.44
1–2	56	97.61			
3–4	94	98.91			
5–8	52	110.37			

* *p* < 0.05.

**Table 5 healthcare-12-00579-t005:** Influencing factors based on the results of the validated questionnaires (*n* = 202).

	Factor	Category	Exp(B)	95% CI for Exp (B)	
				Lower	Upper
**BDI ^1^** <10 points: average scores >10 points: depressive symptoms	Gender	Male (Ref.) Female	2.576	1.296	5.123
**PSS-14 ^2^** <18 points: low stress level >18 points: medium/high stress level	Years worked in healthcare	>20 years (Ref.)			
		0–10 years	7.436	7.111	8.348
		11–20 years	1.830	0.502	6.665
	Gender	Male (Ref.) Female	2.230	1.221	4.073
**STAI-S ^3^** <26 points: low anxiety >26 points: moderate/high anxiety	Gender	Male (Ref.) Female	2.075	1.038	4.146
**STAI-T ^4^** <26 points: low propensity of anxiety >26 points: moderate/high propensity of anxiety	Gender	Male (Ref.) Female	1.995	1.114	3.574

^1^ Beck depression inventory; ^2^ perceived stress scale; ^3^ the state anxiety inventory; ^4^ the trait anxiety inventory.

**Table 6 healthcare-12-00579-t006:** Context analysis of emergency professionals’ perceived stress scale scores (*n* = 202).

**Variables**	**N**	**Mean Rank**	**Sum of Ranks**	**U**	***p*-Value**
Gender				3329.5	<0.001 **
Female	74	120.51	8917.50		
Male	128	93.46	11,962.50		
Workplace				4230.0	0.15
Hungarian Ambulance Service	125	96.84	12,105.50		
Emergency Department	77	109.06	8398.00		
Works night shift				586.0	0.99
Yes	196	101.51	19,896.00		
No	6	101.17	607.00		
Secondary employment				4106.5	0.05
Yes	120	94.72	11,366.50		
No	82	111.42	9136.50		
Regular coffee consumption				2346.0	0.43
Yes	172	102.86	17,692.00		
No	30	93.70	2811.00		
Regular smoking				4500.5	0.35
Yes	80	106.24	8499.50		
No	122	98.39	12,003.50		
Chronic illnesses				4514.0	0.58
Yes	74	98.50	7289.00		
No	128	103.23	13,214.00		
**Variables**	**N**	**Mean Rank**	**Kruskal–Wallis H**	**df**	***p*-value**
Educational attainment			1.446	2	0.49
Elementary school	16	89.63			
Secondary school	102	99.21			
College/university	84	106.55			
Marital status			9.764	3	0.02 *
Single	39	57.03			
Registered partnership	55	54.90			
Married	99	52.03			
Divorced	9	58.80			
Number of children			4.859	2	0.09
Have no children	140	107.52			
1–2	56	88.15			
3–4	6	85.58			
Type of residence			0.564	2	0.75
Large town	48	107.02			
Town	111	99.67			
Village	43	100.07			
Qualifications			15.685	3	0.001 *
Ambulance driver/rescue technician	32	65.66			
Emergency nurse	104	104.45			
Paramedic	48	112.10			
Emergency doctor	18	119.8			
The type of shift			9.603	3	0.02 *
8 h	7	95.50			
12 h	51	122.76			
24 h	79	90.97			
Also, 12 and 24 h	65	98.26			
Working years in the healthcare sector			7.385	3	0.001 *
0–10 years	101	517.81			
11–21 years	58	101.06			
22–30 years	29	85.81			
31–41 years	14	73.96			
Number of treated patients/shift			3.343	3	0.34
1–5	78	95.02			
6–10	64	101.44			
11–20	32	102.55			
>20	28	118.50			
Number of distressing emergency care cases/year			8.324	3	0.04 *
0	10	67.85			
1–10	157	99.07			
11–50	29	119.12			
>50	6	136.00			
Number of leisure activities/person			0.856	2	0.65
1–2	56	97.46			
3–4	94	100.55			
5–8	52	107.58			

* *p* < 0.05; ** *p* < 0.001.

**Table 7 healthcare-12-00579-t007:** Context analysis of emergency professionals’ State and Trait Anxiety Inventory scores (*n* = 202).

STAI-S ^1^					
**Variables**	**N**	**Mean Rank**	**Sum of Ranks**	**U**	***p*-Value**
Gender				3580.0	0.004 *
Female	74	117.12	8667.00		
Male	128	92.47	11,836.00		
Workplace				4145.5	0.10
Hungarian Ambulance Service	125	96.16	12,020.50		
Emergency Department	77	110.16	8482.50		
Works night shift				537.0	0.72
Yes	196	101.24	19,843.00		
No	6	110.00	660.00		
Secondary employment				4709.5	0.61
Yes	120	99.75	11,969.50		
No	82	104.07	8533.50		
Regular coffee consumption				2273.0	0.30
Yes	172	103.28	17,765.00		
No	30	97.33	2738.00		
Regular smoking				4195.0	0.09
Yes	80	110.06	8805.00		
No	122	95.89	11,698.00		
Chronic illnesses				4380.5	0.37
Yes	74	96.70	7155.50		
No	128	104.28	13,347.50		
**Variables**	**N**	**Mean Rank**	**Kruskal–Wallis H**	**df**	***p*-value**
Educational attainment			0.043	2	0.98
Elementary school	16	103.44			
Secondary school	102	100.74			
College/university	84	102.06			
Marital status			9.640	3	0.02 *
Single	39	106.88			
Registered partnership	55	112.64			
Married	99	89.92			
Divorced	9	137.50			
Number of children			11.694	2	0.003 *
Have no children	140	110.85			
1–2	56	80.17			
3–4	6	82.42			
Type of residence			3.025	2	0.22
Large town	48	113.92			
Town	111	96.39			
Village	43	100.84			
Qualifications			6.064	3	0.11
Ambulance driver/rescue technician	32	81.20			
Emergency nurse	104	102.98			
Paramedic	48	104.51			
Emergency doctor	18	121.00			
The type of shift			11.350	3	0.02 *
8 h	7	101.64			
12 h	51	121.51			
24 h	79	86.46			
Also, 12 and 24 h	65	104.07			
Working years in the healthcare sector			5.835	3	0.12
0–10 years	101	110.53			
11–21 years	58	97.30			
22–30 years	29	84.28			
31–41 years	14	89.50			
Number of treated patients/shift			6.719	3	0.08
1–5	78	90.97			
6–10	64	110.28			
11–20	32	94.86			
>20	28	118.34			
Number of distressing emergency care cases/year			8.345	3	0.04 *
0	10	52.75			
1–10	157	102.16			
11–50	29	110.84			
>50	6	120.25			
Number of leisure activities/person			2.132	2	0.34
1–2	56	97.74			
3–4	94	100.33			
5–8	52	110.89			
STAI-T ^2^					
**Variables**	**N**	**Mean Rank**	**Sum of Ranks**	**U**	***p*-value**
Gender				3559.0	0.003 *
Female	74	117.41	8688.00		
Male	128	92.30	11,815.00		
Workplace				4518.5	0.47
Hungarian Ambulance Service	125	99.15	12,393.50		
Emergency Department	77	105.32	8109.50		
Works night shift				535.5	0.71
Yes	196	101.77	19,946.50		
No	6	92.75	556.50		
Secondary employment				4414.5	0.22
Yes	120	97.29	11,674.50		
No	82	107.66	8828.50		
Regular coffee consumption				2455.0	0.67
Yes	172	102.23	17,583.00		
No	30	97.33	2920.00		
Regular smoking				4661.0	0.59
Yes	80	104.24	8339.00		
No	122	99.70	12,164.00		
Chronic illnesses				4706.5	0.94
Yes	74	101.90	7540.50		
No	128	101.27	12,962.50		
**Variables**	**N**	**Mean Rank**	**Kruskal–Wallis H**	**df**	***p*-value**
Educational attainment			0.935	2	0.63
Elementary school	16	111.0			
Secondary school	102	103.36			
College/university	84	97.43			
Marital status			11.333	3	0.01 *
Single	39	110.21			
Registered partnership	55	109.28			
Married	99	89.63			
Divorced	9	146.78			
Number of children			6.432	2	0.04 *
Have no children	140	108.44			
1–2	56	85.98			
3–4	6	84.50			
Type of residence			0.426	2	0.81
Large town	48	104.08			
Town	111	99.08			
Village	43	104.86			
Qualifications			1.951	3	0.58
Ambulance driver/rescue technician	32	89.80			
Emergency nurse	104	102.70			
Paramedic	48	102.70			
Emergency doctor	18	112.19			
The type of shift			9.825	3	0.02 *
8 h	7	85.93			
12 h	51	121.15			
24 h	79	89.15			
Also, 12 and 24 h	65	102.77			
Working years in the healthcare sector			3.419	3	0.33
0–10 years	101	108.13			
11–21 years	58	99.15			
22–30 years	29	90.95			
31–41 years	14	85.29			
Number of treated patients/shift			2.564	3	0.46
1–5	78	94.29			
6–10	64	107.45			
11–20	32	99.30			
>20	28	110.52			
Number of distressing emergency care cases/year			10.214	3	0.02 *
0	10	51.80			
1–10	157	100.90			
11–50	29	119.29			
>50	6	113.92			
Number of leisure activities/person			4.015	2	0.13
1–2	56	92.31			
3–4	94	99.81			
5–8	52	114.44			

* *p* < 0.05; ^1^ The state anxiety inventory; ^2^ the trait anxiety inventory.

## Data Availability

The data presented in this study are available on request from the corresponding author. The datasets generated and/or analyzed during the state study are not publicly available due to privacy, confidentiality, and other restrictions.

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
