# Peer review of "The Occurrence of Anxiety, Depression, and Distress among Professionals Working in Emergency Care"

_healthcare, 2024, doi:10.3390/healthcare12050579_

Round 1

Reviewer 1 Report

Comments and Suggestions for Authors

Thank you very much for having the opportunity to review this paper. This study presents an interesting topic, however, many important questions still need to be resolved in order to consider this manuscript ready to publication:

1. Line 38 says “(p=0.02; OR=2.4, CI 95%=0.7.1-8.3)”; Lines 233-235 says “Women scored higher on both the current level of anxiety (p=0,007; ES=0,22; rs=0,2; OR=2.1, CI 95%=1,0-4,1)...” and Lines 224-225 says “Perceived stress levels decreased proportionally with increasing years spent with work in healthcare (p=0.02; p=0.20; rs=-0.2; OR=2.4, CI 95%=0.7.1-8.3)”.

A 95% confidence interval for the odds ratio also provides a test of the null hypothesis that the odds ratio is 1 at the 5% significance level. If the confidence interval does not include 1, we reject H0 and conclude that the odds for the two groups are different; if the interval does include 1, the data do not provide enough evidence to distinguish the groups in this way. Nonetheless, the interpretation of the obtained p-value leads to the opposite conclusion. In the interpretation of the model, the authors concluded that there is a difference between the groups, however the confidence interval for odds ratio includes the number 1. Please, it’s extremely necessary to review this point.

2. Please, note that it is necessary to replace commas by dots to express decimal numbers throughout the manuscript.

3. Section 2.2 says “The data collection did not involve questions to identify respondents, ensuring their anonymity.” Given that no personal data is collected, what was the procedure followed to ensure that each participating professional completed the questionnaire only once?

4. In Section 2.2, it is suggested that the manuscript include the total number of hospitals that had professionals participating in this study as respondents.

5. Sample size calculation: What is the value of the standard deviation that the researcher took into account when calculating the sample size? Additionally, how did the researcher estimate that standard deviation? That information was not provided in the manuscript.  

6. Sample size calculation: Please, explain the justification for using the N2/N1 allocation ratio of 0.37.

7. Figure 1. It is suggested that this figure be transferred to supplementary material. As it brings no relevant information where it is now placed.

8. Section 2.4: It is suggested that authors indicate the cut-off value used for classification in each questionnaire. For example, for the STAI score, what are the scores in this questionnaire that are considered to be “Low anxiety level”, “Medium anxiety level” and “High anxiety level”. Please, provide bibliographic references that support this classification.

9. The Results section needs improvement. It is proposed to rewrite it. Some points are:

  • Most of the data mentioned in the text is not presented in tables. The reader could get lost in the search for information, as some of the information is given in the text and some comes from your own interpretation of the table. It is suggested that this information be included in the tables. In the text, the authors could highlight the main results obtained. If the table becomes too large, it could be included in the supplementary material.

  • The text mentions the percentage of widows, but this class has not been included in the table. 

  • The current format of Tables 1 and 2 is not easy to understand. It is suggested to reorganize the presentation, perhaps considering the classes of variables as rows rather than columns. There is no need for repetition of the word 'persons' in all cells.

  • Lines 150-152 says “Among women, there is a higher proportion of college and university graduates than among men”. Is this statement based on the result of the hypothesis test? It is suggested that authors perform hypothesis tests to compare demographic data between men and women. 

  • Min, Max and Mean are sensitive to outliers. It is suggested that the authors include other measures that are not sensitive as quartiles (Q1, Q2 and Q3).

  • It is suggested that the sentence in lines 173-174 be rewritten as follows “Among the participants, 56 persons (27,7%) are engaged regularly in one or two free-time activities, 96 persons (47,5%) in three or four free-time activities, and 50 persons (24,8%) in more than four types of free-time activities.”  and the sentence in lines 138-139 as follows “... and most participants are middle-aged (ranged between 18 and 63 years)”.

10. Table 2. The variable “Number of fostered children” was presented but not the variable relating to the total number of children. Was this information collected?

11. Was it possible to measure the workload of housework and childcare with the questionnaires used? In the authors' opinion, could  the greater workload of women in domestic work and childcare as a cause of the highest percentile of stress and depression among women (compared to men) be an explanation for the results? It is suggested that authors mention articles and discuss this viewpoint.

12. It is suggested that the authors present an analysis of coffee consumption and tabagism by gender. Were data about alcohol consumption also collected?

13. A table or graph showing the most common leisure activities by gender might be interesting. Since, if the women are overburdened with housework and childcare, it is possible that the percentile of women that are engaged in leisure activities will be lower than that of men.

14. In this sense, a table or a graph showing the most common leisure activities by depression / stress / anxiety class might be interesting to confirm the statement made in the Conclusion section: “According to our results, promoting leisure activities and facilitating attendance in programs by the employer organization could benefit emergency care. Engaging in these activities helps to reduce the frequency of harmful addictions and the level of anxiety.” 

15.The ES measure (Cramer's association coefficient) is calculated by the authors in many cases. However, no interpretation of these results was made. The authors must discuss the results obtained. In addition, it is suggested that the authors report the test statistics obtained in addition to the p-value. This modification will allow the reader to identify the type of test that is used in each case.

16. In lines 213-215 the ES measure has been calculated, but it seems that these variables are quantitative. Please, clarify.

17. It is not clear when the logistic model has been in use. It is suggested that the authors present the results of this model in a table showing the coefficients, OR, p-value and other measures.

Comments on the Quality of English Language

Moderate editing of English language is required.

Reviewer 2 Report

Comments and Suggestions for Authors

The introduction needs to be clearer, especially from line 61 onwards. Some reference to a previous article is needed to explain why this needs to be investigated.

Current and well expressed references.

The methodology is perfectly expressed.

Results expressed clearly.

The discussion can be improved by comparing the results found with articles that have the same population as yours. On line 273 it is compared to students. It would also be advisable to include some relaxation techniques apart from the support of mental services.

Reviewer 3 Report

Comments and Suggestions for Authors

In detail, I read and analysed the manuscript The Occurrence of Anxiety, Depression, and Distress Among Professionals Working in Emergency Care.

The topic is up to date because of the high prevalence of anxiety, depression, and distress among Emergency Care workers. However, the manuscript has several weaknesses, and it is recommended that the authors make certain corrections.

Introduction - Namely, the introduction is not fully in line with the problem being investigated. If the incidence of anxiety, depression and distress is examined, the authors should focus more on why it is important to research it for professionals working in emergency care. What are the findings so far?

Also, it is unclear why the authors mention two study phases. Why is it necessary when the results of the second phase are not revealed, nor are factors such as a sense of coherence, social support, sleeping habits, and others explained in the introduction? Certainly, the manuscript was not reported as a preliminary communication.

It is necessary to clearly state only the study’s objective whose results are being addressed.

In line 45, exclude the word “our because there is no other world.

Material and Methods - Stick to one term, e.g. participants

Considering that the questionnaires are distributed through several channels, it is necessary to state how they are controlled so that one person does not complete the questionnaire more than once. The sentence in lines 87-88 states that the link to the questionnaire was distributed, not the questionnaire itself.

Part of the sentence in line 30, “higher scores represented greater degrees of depressed mood„, should be excluded because the above text explains the questionnaire scoring.

In the part where the authors describe the PSS-14 and STAI questionnaires, the term “questions” should be replaced by item items (lines 129 and 139.

It is necessary to explain what a self-edited questionnaire means. Is it a general questionnaire?

Results

In line 149. Majority (lowercase required). In line 152, the authors state, “in cities,” and in the table, it is a large town, town, and village. With that, it is necessary to synchronise the dates.

All in all, the results are too extensive. A lot of general variables are shown that are not further investigated. Instead, it is necessary to show only sociodemographic variables and results that answer the set goals in more detail (table 5 shows only gender and years of work experience). Everything else is displayed textually and hard to follow. Focus more on inferential statistics and not just on descriptive presentation.

The results described on page 8 should be shown in tabular form, and the sociodemographic characteristics should show only those researched and commented on in the discussion.

Discussion - inadequate to the goal and the results presented. The authors do not analyse the results of the occurrence of anxiety, depression, and distress.

Comments on the Quality of English Language

Round 2

Reviewer 1 Report

Comments and Suggestions for Authors

I would like to thank the authors for answering my questions, accepting the main suggestions and making the needed changes in the manuscript. 

Therefore, to avoid any misunderstanding among readers regarding CI and p-values (see my note 1 below), it is recommended that the authors present the results in the Abstract and text in the Results section using two decimal places.

Note 1: A 95% confidence interval for the odds ratio also provides a test of the null hypothesis that the odds ratio is 1 at the 5% significance level. If the confidence interval does not include 1, we reject H0 and conclude that the odds for the two groups are different; if the interval does include 1, the data do not provide enough evidence to distinguish the groups in this way. Nonetheless, the interpretation of the obtained p-value leads to the opposite conclusion. In the interpretation of the model, the authors concluded that there is a difference between the groups, however the confidence interval for odds ratio includes the number 1. Please, it’s extremely necessary to review this point.

In addition, it is recommended that the authors review the manuscript once more for fine adjustments. For example:

a. Check the values presented in Abstract and in the text as there are some inconsistencies. An example,

  • Abstract: depressive symptoms (p=0.009; OR=2.6, 95% CI=1.3-5.1);

  • Results section: In our study investigating the risk factors associated with the appearance of depressive symptoms, logistic regression analysis revealed that gender was a significant predictor (p=0.007; OR=2.6, 95% CI=1.3-5.1).

b. The lines 499-508 and 517-526 are the same (duplicated).

Reviewer 3 Report

Comments and Suggestions for Authors

I read and carefully analyzed the revised version of the manuscript The occurrence of anxiety, depression, and distress among professionals working in emergency care.

However, the authors have corrected almost all suggestions except for one, the most important one, which is the presentation of the characteristics of the sample.

If the study aims to assess the frequency of anxiety, depression, and distress, then it should be dealt with, and the results should be presented. Such a detailed presentation and analysis of the characteristics of the samples has no scientific contribution.

I agree that they can be a behavioural indicator of anxiety, depression, and distress, but in that case,  the results from item 3.3. The results of the validated questionnaires should be presented tabularly, not textually, because it is very difficult for readers to follow.
